# *Lacticaseibacillus rhamnosus* MG4706 Suppresses Periodontitis in Osteoclasts, Inflammation-Inducing Cells, and Ligature-Induced Rats

**DOI:** 10.3390/nu14224869

**Published:** 2022-11-17

**Authors:** Seonyoung Kim, Ji Yeon Lee, Jeong-Yong Park, YongGyeong Kim, Chang-Ho Kang

**Affiliations:** MEDIOGEN Co., Ltd., Biovalley 1-ro, Jecheon-si 27159, Republic of Korea

**Keywords:** lactic acid bacteria, anti-inflammatory, anti-osteoclastogenesis, periodontitis, RANKL/OPG system

## Abstract

Periodontitis is a chronic inflammatory disease characterized by tooth loss due to inflammation and the loss of alveolar bone. Periodontitis is closely related to various systemic diseases and is emerging as a global health problem. In this study, we investigated the anti-inflammatory effect of lactic acid bacteria (LAB) in vitro on *Porphyromonas gingivalis* (*P. gingivalis*) LPS-activated RAW264.7 and human gingival fibroblasts-1 (HGF-1) cells and the anti-osteoclastogenic effect of LAB on RANKL-induced RAW264.7 cells. All LAB strains (*Lacticaseibacillus rhamnosus* MG4706, MG4709, and MG4711) inhibited nitric oxide (NO)/*inducible nitric oxide synthase* (*iNOS*) in *P. gingivalis* LPS-activated RAW264.7 cells and pro-inflammatory cytokines (*IL-1β* and *IL-6*) and matrix metalloproteinase (*MMP-8* and *MMP-9*) in HGF-1 cells. In addition, LAB treatment inhibited osteoclastogenesis by reducing tartrate-resistant acid phosphatase (TRAP) activity and *cathepsin K* (*CtsK*) through the downregulation of *nuclear factor of activated T cells cytoplasmic 1* (*NFATc1*) and *c-fos* gene expression in RANKL-induced RAW264.7 cells. Administration of MG4706 alleviated alveolar bone loss indices and reduced the gene expression of *IL-1β*, *IL-6*, *MMP-8, MMP-9,* and *RANKL*/*OPG* ratio in gingival tissue. In conclusion, *L. rhamnosus* MG4706 has the potential to alleviate periodontitis.

## 1. Introduction

Periodontitis is a chronic inflammatory disease that causes tooth loss due to the destruction of alveolar bone and inflammation of the gums [1,2]. Periodontitis is closely related to various systemic diseases, such as cardiovascular disease and arthritis, and is a major public health problem affecting all populations worldwide [3,4]. Although several gram-negative anaerobic bacteria, such as *Fusobacterium nucleatum*, *Treponema denticola*, and *Aggregatibacter actinomycetemcomitans*, are involved in the pathogenesis of this disease, *Porphyromonas gingivalis* (*P. gingivalis*) is considered the main etiological agent [5,6]. Lipopolysaccharide (LPS) in the *P. gingivalis* membrane is an inducer of nitric oxide (NO) and inflammatory cytokines, such as interleukin (IL)-1β and IL-6, which promote the receptor activators of nuclear factor κB (NF-κB) ligand (RANKL) signaling pathway, leading to the destruction of connective tissue, including alveolar bone resorption [1,3,7,8]. NO is produced from inducible nitric oxide synthase (iNOS) and causes bone loss by regulating proinflammatory cytokine expression. As mercaptoethyl guanidine, one of the iNOS inhibitors, reduces bone loss, NO is used as a pharmaceutical target in periodontal disease management [9]. The expression of IL-1β and IL-6 is associated with sustained tissue destruction, and these cytokines promote the destruction of alveolar bone and extracellular matrix by periodontitis [10]. RANKL is a critical factor for osteoclastogenesis, and it induces osteoclast maturation and activity by binding to RANK, a receptor present on the surface of osteoclast precursors [11]. In addition, matrix metalloproteinases (MMPs) that decompose the constituting components of the periodontal tissue are generated excessively, leading to increased osteoclast activity and further progress of periodontitis [12].

Periodontal disease can be prevented by removing tartar and plaque through brushing and scaling; however, these procedures have limitations [3]. In addition, the use of doxycycline to treat periodontitis has the disadvantage of producing side effects such as photosensitivity, nausea, and hyperpigmentation [13,14]. Therefore, a new method that differs from mechanical methods is required for the prevention and treatment of periodontal disease. One of the new strategies that are gaining significant interest is the use of probiotics to interfere with the interactions between the host cells and pathogens and restore homeostasis among the microbial community in host cells [15,16].

Probiotics are defined as living microorganisms that can exert beneficial effects on the host when ingested in appropriate amounts. Probiotics are mostly composed of lactic acid bacteria (LAB), including *Lactobacillus*, *Bifidobacterium*, *Streptococcus*, and *Enterococcus*. Probiotics principally maintain the health of the host by maintaining the balance between beneficial strains, secretion of antibacterial substances, and immune regulation [17]. In addition, LAB have been actively studied for their effects, such as anti-obesity [18,19,20], anti-diabetes [21,22,23], and anti-atopic dermatitis [24,25]. However, studies on the effect of probiotics on periodontitis are lacking.

Therefore, we confirmed the anti-inflammatory effect of LAB in *P. gingivalis* LPS-activated RAW264.7 and human gingival fibroblasts-1 (HGF-1) cells and elucidated the anti-osteoclastogenic effect exerted in vitro by LAB in RANKL-induced RAW264.7 cells. In addition, the periodontitis relief efficacy of LAB was evaluated in vivo using a rat model of ligation-induced periodontitis.

## 2. Materials and Methods

### 2.1. Preparation of Experiment Sample

The three LAB strains (*Lacticaseibacillus rhamnosus* MG4706, MG4709, and MG4711) supplied by MEDIOGEN (Jecheon, Korea) were isolated from the oral cavities of healthy adults. The strains were cultured in de Man, Rogosa, and Sharpe (MRS) broth (BD Difco, Franklin Lakes, NJ, USA) for 18 h at 37 °C and adjusted to a density of 10^8^–10^9^ CFU/mL. The adjusted culture medium was inoculated with 2% in fresh MRS broth and further cultured for 24 h. Then, the culture medium was centrifuged at 4000 rpm for 5 min and filtered to obtain a cell-free supernatant (CFS). The CFS was stored at −80 °C until used in further experiments, and a part of it was freeze-dried (CFSP).

### 2.2. Cell Culture and Differentiation

RAW 264.7 (murine macrophages) and HGF-1 (ATCC, Manassas, VA, USA) cells were cultured in Dulbecco’s Modified Eagle Medium (DMEM; Gibco, Grand Island, NY, USA) containing 10% heat-inactivated fetal bovine serum (FBS; Gibco) and 100 units/mL of penicillin/streptomycin (P/S; Gibco) at 37 °C and under 5% CO_2_. When RAW 264.7 cells differentiated into osteoclasts, the medium was replaced with minimum essential medium α (MEMα, Gibco) containing 100 ng/mL RANKL (R&D Systems, Minneapolis, MN, USA) for 2–4 days.

### 2.3. Nitric Oxide (NO) Production

NO production was analyzed according to the method described by Kim et al. with some modifications [26]. RAW 264.7 cells were seeded in 96-well plates at a concentration of 2 × 10^5^ cells/well and incubated overnight at 37 °C. The cells were treated with DMEM (FBS-free) containing 5 μg/mL of *P. gingivalis* LPS (PgLPS; InvivoGen, San Diego, CA, USA) and 5% CFS and further incubated for 36 h. Then, the supernatant (100 µL) and fresh Griess reagent (100 μL) were mixed in a new plate and incubated at 25 °C for 10 min. The absorbance was measured at 550 nm using a microplate reader (BioTek, Winooski, VT, USA). NaNO_2_ was used as the standard to quantify NO. Cytotoxicity was determined using the MTT method [27].

### 2.4. Tartrate-Resistant Acid Phosphatase (TRAP) Activity

TRAP activity was measured using a modified version of the assay described by Kim et al. [28]. RAW 264.7 were seeded in 96-well plates (1 × 10^4^ cells/well) and incubated overnight at 37 °C. The cells were treated with MEMα containing 100 ng/mL RANKL and 400 μg/mL CFSP for 4 days, and the medium was replaced with fresh medium every 2 days. Then, the cells were lysed using 0.05% Triton X-100 in saline. The lysate was mixed with an equal volume of TRAP working solution (10 mM p-nitrophenyl phosphate in 50 mM citrate buffer containing 10 mM sodium tartrate, pH 4.7) and incubated for 1 h at 37 °C. The absorbance was determined at 405 nm using a microplate reader (BioTek).

### 2.5. Quantitative Real Time-Polymerase Chain Reaction (qRT-PCR)

Gene expression was confirmed using qRT-PCR according to the method described by Lee et al. [29]. RAW 264.7 cells were treated with CFS and PgLPS for 3 h to identify the inflammatory genes (*iNOS*) and with CFSP and RANKL for 48 h to identify osteoclastogenesis-related genes (*NFATc1*, *c-fos*, and *CtsK*). HGF-1 cells were pre-treated with PgLPS for 6 h, followed by treatment with CFSP for 24 h to identify inflammatory genes (*IL-1β* and *IL-6*), *MMP-8*, and *MMP-9*. The gingival tissue was collected at the end of the animal experiment, and identification of the inflammatory and alveolar bone loss-related genes was performed (*IL-1β*, *IL-6*, *MMP-8*, *MMP-9*, *RANKL*, and *OPG*). Total RNA from cells was isolated using the NucleoZOL reagent (Macherey-Nagel, Düren, Germany). The cDNA was synthesized using a reverse transcriptase premix (iNtron, Seongnam, Korea), and gene expression was determined using iQ SYBR Green Supermix (Bio-Rad, Hercules, CA, USA). qRT-PCR was performed using the CFX Connect Real-Time PCR Detection System (Bio-Rad). The PCR reaction was pre-cycling at 95 °C for 3 min, followed by 39 cycles of denaturation (95 °C, 10 s), annealing (55–65 °C, 10 s), and extension (72 °C, 30 s). Relative target gene expression was normalized to that of *GAPDH* and calculated using the 2^−ΔΔCT^ method. The primer sequences used for this analysis are listed in Table 1.

### 2.6. Aninal Experiment Design

Sprague–Dawley (SD) rats (male, 6 weeks; Koatech, Pyeongtaek, Korea) were maintained at a relative humidity of 55 ± 15% and a temperature of 23 ± 3 °C with a light-dark cycle of 12 h. All animal experiments were approved by the Institutional Animal Care and Use Committee (KNOTUS 22-KE-0228). After a week of acclimatization, the rats were randomly assigned to three groups: (1) normal group (no ligation and vehicle, *n* = 8), (2) ligation group (ligation and vehicle, *n* = 8), and (3) MG4706 group (ligation and *L. rhamnosus* MG4706 (1 × 10^9^ CFU/day), *n* = 8). Periodontitis was induced by ligating the right second molar of the mandible using a 4-0 silk suture. After oral administration of the sample (vehicle or MG4706) and induction of ligation for eight weeks, gingival tissues were collected and used for other analyses. For the duration of the experiment, the rats were given a standard diet (Cargill Inc., Seongnam, Korea) and free access to purified water.

### 2.7. Micro-Computerized Tomography (Micro-CT)

Micro-CT analysis was performed according to a previously described method [30]. The cemento-enamel junction (CEJ)-alveolar bone crest (ABC) distance and furcation involvement were used as alveolar bone loss indices.

### 2.8. Statistical Analysis

All experiments were performed in triplicate and are presented as the mean ± standard error of the mean (SEM). Statistical analysis was performed using GraphPad Prism (version 5; GraphPad Inc., San Diego, CA, USA), and Dunnett’s test was used to determine the significance of differences (*p* < 0.05) between the control and sample groups.

## 3. Results

### 3.1. Effect of LAB on the NO Production and Gene Expression of iNOS in PgLPS-Activated RAW264.7 Cells

NO production and *iNOS* mRNA expression were investigated in PgLPS-activated RAW264.7 cells treated with LAB CFS. PgLPS increased the production of NO to 14 μM, but all LAB CFSs significantly decreased NO production by 6.8–7.7 μM (inhibition rate, 45–51%) (Figure 1A). Similarly, *iNOS* gene expression was increased by PgLPS and was significantly reduced by all LAB strains (Figure 1B). Cytotoxicity was not observed in more than 85% of the treated groups (Figure 1C).

### 3.2. Effect of LAB on TRAP Activity and Osteoclastogenesis Specific Gene Expression in RANKL-Induced RAW264.7 Cells

TRAP activity is an indicator of osteoclast differentiation and is closely related to increased bone resorption. TRAP activity was increased by RANKL treatment and was significantly decreased by treatment with MG4706, MG4709, and MG4711 strains. Cytotoxicity was not observed in more than 85% of all treated groups (Figure 2A).

The effect of LAB on the expression of the genes involved in osteoclastogenesis, such as *NFATc1*, *c-fos*, and *CtsK*, was investigated using qRT-PCR (Figure 2B). *NFATc1*, *c-fos*, and *CtsK* gene expression levels were significantly higher in the RANKL-treated group than in the control group without RANKL. In RANKL-induced RAW264.7 cells, *NFATc1*, *c-fos*, and *CtsK* gene expression levels were reduced compared with that of RANKL only-treated group on treatment with CFSP of MG4706 (0.62-, 0.30-, and 0.45-fold lower, respectively), MG4709 (0.89-, 0.80-, and 0.61-fold lower, respectively), and MG4711 (0.38-, 0.24-, and 0.22-fold lower, respectively).

### 3.3. Effect of LAB on Pro-Inflammatory Cytokines and MMP Gene Expression in PgLPS-Stimulated HGF-1 Cells

The effects of LAB treatment on pro-inflammatory cytokine and MMP gene expression were assessed in PgLPS-stimulated HGF-1 cells. The expression levels of *IL-1β* and *IL-6* genes were increased by PgLPS treatment, whereas LAB treatment significantly reduced *IL-1β* and *IL-6* expression (Figure 3A,B). *MMP-8* gene expression increased following PgLPS treatment but significantly decreased compared with that of the PgLPS only treated-group following treatment with LAB strains MG4706 (0.42-fold decrease), MG4709 (0.46-fold decrease), and MG4711 (0.62-fold decrease) (Figure 3C). In addition, *MMP-9* gene expression was significantly downregulated compared with that of the PgLPS only treated-group on treatment with MG4706 (0.59-fold decrease), MG4709 (0.60-fold decrease), and MG4711 (0.60-fold decrease) strains (Figure 3D).

### 3.4. Effect of L. rhamnosus MG4706 on Alveolar Bone Loss in Ligatured-induced Periodontal Rats

In our in vitro study, *L. rhamnosus* MG4706, which had a higher production yield than that of *L. rhamnosus* MG4711 (data not shown), was selected for in vivo experiments. Since the loss of alveolar bone worsened as periodontitis progressed, we analyzed the cemento-enamel junction (CEJ)–alveolar bone crest (ABC) distance and the involvement of furcation induced by ligation in the rats. CEJ–ABC distance and furcation involvement increased in the ligation group, whereas both parameters were significantly downregulated in the MG4706 group (Figure 4).

### 3.5. Effect of L. rhamnosus MG4706 on Pro-Inflammatory Cytokines and MMP Gene Expression in Gingival Tissue

The effects of *L. rhamnosus* MG4706 administration on the expression of pro-inflammatory cytokines (*IL-1β* and *IL-6*) and MMPs (*MMP-8* and *MMP-9*) in gingival tissues were evaluated using qRT-PCR (Figure 5). The *IL-1β, IL-6, MMP-8*, and *MMP-9* gene levels in the ligation group were higher than that in the normal group. The MG4706 group showed 0.62-, 0.66-, 0.92-, and 0.64-fold reduced expression of *IL-1β, IL-6, MMP-8*, and *MMP-9* than that in the ligation group, respectively.

### 3.6. Effect of L. rhamnosus MG4706 on Alveolar Bone Loss Related Gene Expression in Gingival Tissue

The effects of *L. rhamnosus* MG4706 administration on the expression of alveolar bone loss-related genes, such as *RANKL* and *osteoprotegerin (OPG*), in gingival tissue were evaluated using qRT-PCR. The expression level of *RANKL* showed 0.38-fold significant decrease in the MG4706 group than that in the ligation group (Figure 6A), whereas the expression level of *OPG* was 1.26-fold higher than that in the ligation group (Figure 6B). The *RANKL*/*OPG* ratio was increased by ligation and was significantly downregulated by the administration of MG4706 (Figure 6C).

## 4. Discussion

Periodontal disease is a global health problem due to its negative impact on quality of life and a high incidence among the elderly, young, and middle-aged individuals [3,8]. Periodontitis is a chronic inflammatory disease that progresses due to continuous inflammation incited by oral pathogens, resulting in tooth loss caused by the destruction of alveolar bone and gingival tissue [31]. Individuals affected by this disease have a significantly higher risk of developing systemic diseases such as atherosclerosis and stroke; therefore, early prevention is necessary [32]. The diverse microbiota of the human oral cavity contributes to the defense against external pathogens and the development of the host’s immune system by balancing the different microorganism ratios. However, an imbalance in the oral microbiome, such as an increase in the proportion of anaerobic gram-negative bacteria, leads to periodontitis [33,34]. Therefore, the study of probiotics, such as LAB, is necessary for restoring microbial balance. We confirmed the anti-periodontitis effect of LAB isolated from the oral cavity of healthy adults in PgLPS-activated RAW264.7 and HGF-1 cells and RANKL-induced RAW264.7 cells.

*P. gingivalis* induces the production of NO and inflammatory cytokines such as IL-1β and IL-6 through the activation of the Toll-like receptor (TLR) and causes an inflammatory response. NO is produced by iNOS, and excessive NO production triggers an apoptotic response in the teeth and gingival tissues [35]. Therefore, suppressing the inflammatory response caused by PgLPS may be an effective method to alleviate periodontitis. In our study, *L. rhamnosus* MG4706, MG4709, and MG4711 strains inhibited *iNOS* expression and NO production in PgLPS-activated RAW264.7 cells (Figure 1). In addition, all LAB strains showed downregulated expression of *IL-1β* and *IL-6* genes in PgLPS-stimulated HGF-1 cells (Figure 3A,B). Luan et al. [36] reported that *Leuconostoc mesenteroides* LVBH107 suppressed NO production by reducing *iNOS* mRNA expression in PgLPS-induced RAW264.7 cells and downregulated the mRNA expression of *IL-1β* and *IL-6*.

MMPs are one of the mediators that play an important role in the progression of periodontitis. Periodontal tissue degradation and destruction in periodontitis are associated with excessive secretion of MMPs [37,38,39]. Stimulation by *P. gingivalis* induced large amounts of MMP-8 and -9 secretion in whole blood [40]. MMP-8 degrades type I, II, and III collagen, and MMP-9 degrades type IV and V collagen and other extracellular matrix proteins. Both enzymes are closely related to periodontal tissue destruction and are the major MMPs most frequently expressed in the gingival tissue and oral fluid of patients with periodontitis [6,41]. In this study, the three LAB strains downregulated the expression of *MMP-8* and *MMP-9* genes compared with that in the PgLPS only treated-group (Figure 3C,D). Thus, the LAB strains used in this study exhibited anti-inflammatory effects by inhibiting the expression of NO/*iNOS*, pro-inflammatory cytokines, *MMP-8*, and *MMP-9* in RAW264.7 and HGF-1 cells stimulated with PgLPS.

Bone resorption by osteoclasts is a typical symptom of periodontitis. Osteoclasts are derived from monocytes/macrophages and differentiate into mature cells by the involvement of RANKL [42,43]. Mature osteoclasts induce osteoclastogenesis by activating NFATc1 and c-fos and producing osteoclast-specific enzymes such as TRAP and *CtsK* [44,45]. In our study, *L. rhamnosus* MG4706 and MG4711 significantly suppressed TRAP activity and expression of *CtsK*, *NFATc1*, and *c-fos* (Figure 2). Thus, MG4706 and MG4711 exerted anti-osteoclastogenesis effects by reducing TRAP activity and *CtsK* expression by downregulating the expression of *NFATc1* and *c-fos* genes in RANKL-induced RAW264.7 cells. In addition, *L. rhamnosus* MG4706 showed a higher production yield than did *L. rhamnosus* MG4711 (data not shown). Based on these results, *L. rhamnosus* MG4706 was selected for the in vivo experiments in a rat model of ligation-induced periodontitis.

Due to the human-like anatomical features in rat molars, the rat model of ligation-induced periodontitis has been extensively used [26]. Periodontitis may be induced through either local injection of PgLPS or oral administration of bacteria; however, the ligation method of inducing periodontal tissue destruction is faster [46]. Ligation promotes the invasion of bacteria into the connective tissue by inducing the accumulation of plaque around the ligation site and the teeth, causing inflammation and destruction of the alveolar bone [47,48]. Furcation involvement is very common in patients with periodontitis, and the CEJ–ABC distance estimation is the most widely used method to assess periodontitis progression [49,50,51]. In our experiment, *L. rhamnosus* MG4706 administration for eight weeks significantly reduced the CEJ–ABC distance and furcation involvement compared with that in the ligation group (Figure 4), demonstrating the suppression of periodontitis progression by decreasing alveolar bone loss. In addition, analysis of gene expression in the gingival tissue with ligature induced-inflammation showed that the expression of inflammatory cytokines *IL-1β* and *IL-6* was suppressed in the MG4706 group, and the expression of *MMP-9* was also significantly decreased (Figure 5).

RANKL is mainly produced by stromal cells and osteoblasts and is an important factor in osteoclastogenesis. OPG is a decoy receptor of RANKL that inhibits osteoclastogenesis by interfering with the RANKL–RANK interaction [26,52]. It has been reported that the RANKL/OPG system is closely related to the regulation of osteoclast-forming capacity and bone resorption activity in periodontitis [53,54]. Therefore, an imbalance in RANKL/OPG ratio affects the development of periodontitis. Evaluation of the effect of MG4706 on the RANKL/OPG system showed that administration of MG4706 did not enhance *OPG* expression but significantly suppressed that of *RANKL* (Figure 6). Thus, *L. rhamnosus* MG4706 down-regulates the *RANKL*/*OPG* ratio compared with that in the ligation group.

## 5. Conclusions

Our study suggested that LAB treatment inhibited NO/*iNOS* expression in PgLPS-activated RAW264.7 cells and suppressed the gene expression of pro-inflammatory cytokines (*IL-1β* and *IL-6*), *MMP-8*, and *MMP-9* in PgLPS-stimulated HGF-1 cells. In addition, LAB treatment resulted in anti-osteoclastogenesis by reducing TRAP activity and *CtsK* expression through the downregulation of *NFATc1* and *c-fos* gene expression in RANKL-induced RAW264.7 cells. The effects of *L. rhamnosus* MG4706 (selected based on the production yield among the three LAB strains) were evaluated in a rat model of ligature-induced periodontitis. The administration of MG4706 significantly suppressed alveolar bone loss and *IL-1β, IL-6, MMP-8*, and *MMP-9* gene expression in gingival tissues and significantly downregulated the *RANKL*/*OPG* ratio. Thus, MG4706 has the potential to alleviate periodontal disease.

## Figures and Tables

**Figure 1 nutrients-14-04869-f001:**
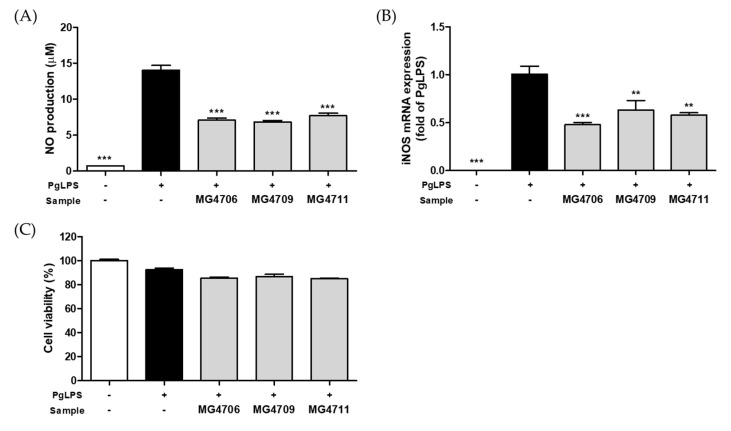
Effect of cell-free supernatant (CFS) from lactic acid bacteria (LAB) on (**A**) nitric oxide (NO) production, (**B**) *inducible nitric oxide synthase* (*iNOS*) mRNA expression, and (**C**) cytotoxicity in *P. gingivalis* LPS (PgLPS)-activated RAW264.7 cells. Data are represented as mean ± SEM (*n* = 3). Significance was analyzed using Dunnett’s test. ** *p* < 0.01, *** *p* < 0.001 versus only treated-PgLPS group.

**Figure 2 nutrients-14-04869-f002:**
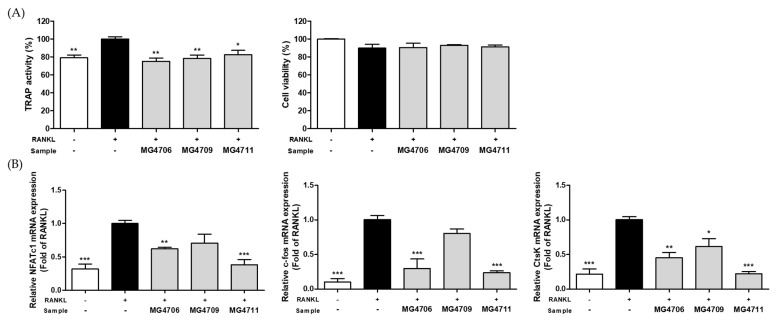
Effect of CFSP from LAB on (**A**) TRAP activity and cytotoxicity and (**B**) expression of osteoclastogenesis specific genes (*NFATc1*, *c-fos*, and *CtsK*) in RANKL-induced RAW264.7 cells. Data are represented as mean ± SEM (*n* = 3). Significance was analyzed by Dunnett’s test. * *p* < 0.05, ** *p* < 0.01, *** *p* < 0.001 versus only treated-RANKL group.

**Figure 3 nutrients-14-04869-f003:**
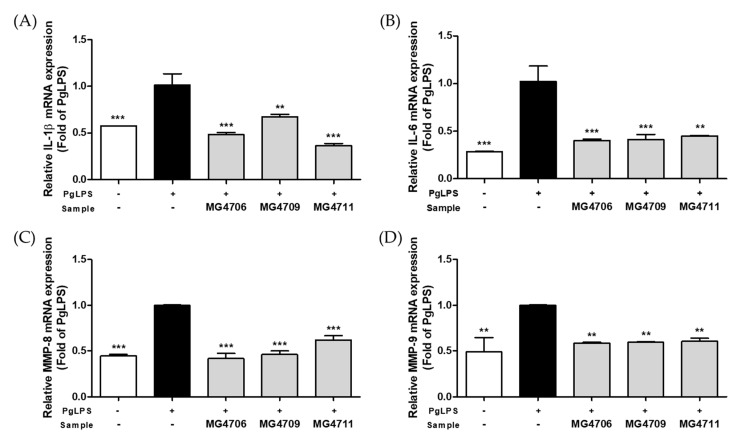
Effect of CFSP from LAB on mRNA expression of (**A**) *interleukin* (*IL*)*-1β* (**B**) *IL-6* (**C**) *matrix metalloproteinase* (*MMP*)*-8* and (**D**) *MMP-9* in PgLPS-stimulated human gingival fibroblasts-1 (HGF)-1 cells. Data are represented as mean ± SEM (*n* = 3). Significance was analyzed by Dunnett’s test. ** *p* < 0.01, *** *p* < 0.001 versus only treated-PgLPS group.

**Figure 4 nutrients-14-04869-f004:**
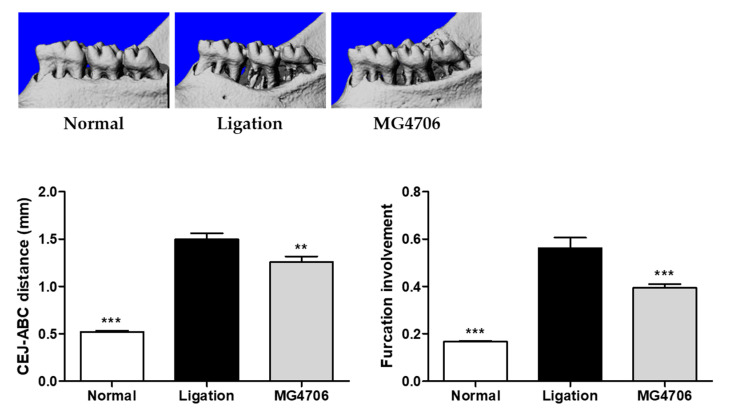
Effect of MG4706 on alveolar bone loss in ligature-induced periodontal rats. The cementoenamel junction–alveolar bone crest (CEJ-ABC) distance and furcation involvement were used as indices of alveolar bone loss and were measured through micro-CT imaging. Data are represented as mean ± SEM (*n* = 8). Significance was analyzed using Dunnett’s test. ** *p* < 0.01, *** *p* < 0.001 versus ligation group.

**Figure 5 nutrients-14-04869-f005:**
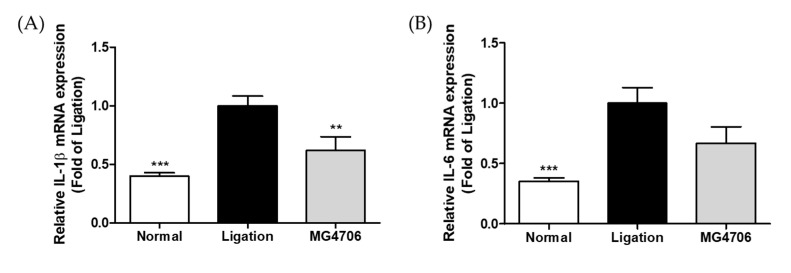
Effect of MG4706 on mRNA expression of (**A**) *IL-1β*, (**B**) *IL-6*, (**C**) *MMP-8*, and (**D**) *MMP-9* in gingival tissue for ligatured-induced periodontal rats. Data are represented as mean ± SEM (*n* = 8). Significance was analyzed using Dunnett’s test. ** *p* < 0.01, *** *p* < 0.001 versus ligation group.

**Figure 6 nutrients-14-04869-f006:**
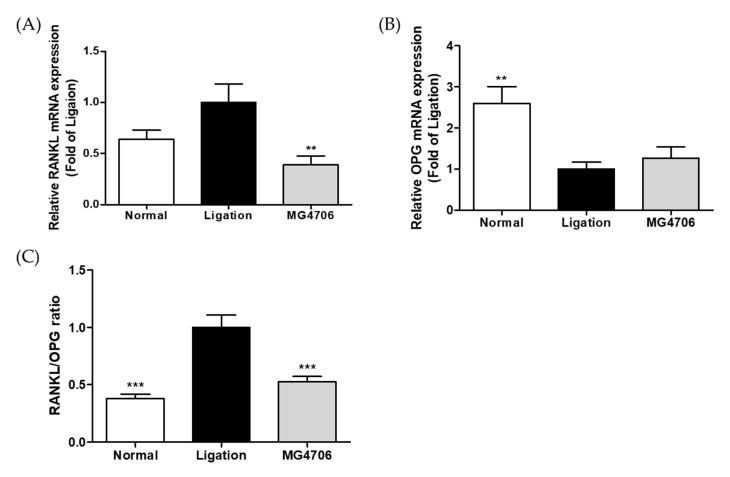
Effect of MG4706 on mRNA expression of (**A**) *RANKL*, (**B**) *OPG*, and (**C**) *RANKL*/*OPG* ratio in gingival tissue for ligatured-induced periodontal rats. Data are represented as mean ± SEM (*n* = 8). Significance was analyzed using Dunnett’s test. ** *p* < 0.01, *** *p* < 0.001 versus ligation group.

**Table 1 nutrients-14-04869-t001:** Primer sequences used for qRT-PCR.

Gene *		Origin	Sequence (5′→3′)	
			Forward	Reverse
Inflammatory	*iNOS*	Mouse	CCTCACGCTTGGGTCTTGTT	GCACAAGGGGTTTTCTTCACG
	*IL-1β*	Human	AGAAGTACCTGAGCTCGCCA	CCTGAAGCCCTTGCTGTAGT
		Rat	TGAAGCAGCTATGGCAACTG	GGGTCCGTCAACTTCAAAGA
	*IL-6*	Human	AAGCCAGAGCTGTGCAGATGAGTA	TGTCCTGCAGCCACTGGTTC
		Rat	TGATGGATGCTTCCAAACTG	GAGCATTGGAAGTTGGGGTA
	*MMP-8*	Human	CCACTTTCAGAATGTTGAAGGGAAG	TCACGGAGGACAGGTAGAATGGA
		Rat	GGATTCCCAAGGAGTGTCCA	CTGGGAACACGCTTGCTATG
	*MMP-9*	Human	GCACGACGTCTTCCAGTACC	GCACTGCAGGATGTCATAGGT
		Rat	GCGCTGGGCTTAGATCATTC	TGGGACACATAGTGGGAGGA
Osteoclastogenesis related	*NFATc1*	Mouse	GGAGAGTCCGAGAATCGAGAT	TTGCAGCTAGGAAGTACGTCT
	*c-fos*	Mouse	CCAGTCAAGAGCATCAGCAA	AAGTAGTGCAGCCCGGAGTA
	*CtsK*	Mouse	GAAGAAGACTCACCAGAAGCAG	TCCAGGTTATGGGCAGAGATT
Alveolar bone loss related	*RANKL*	Rat	CATGAAACCTCAGGGAGCGT	GTTGGACACCTGGACGCTAA
	*OPG*	Rat	GTTCTTGCACAGCTTCACCA	AAACAGCCCAGTGACCATTC
Housekeeping	*GAPDH*	Mouse	TCTCCCTCACAATTTCCATCC	GGGTGCAGCGAACTTTATTG
		Human	ACCCACTCCTCCACCTTTG	CTCTTGTGCTCTTGCTGGG
		Rat	CAAGTTCAACGGCACAGTCAAG	ACATACTCAGCACCAGCATCAC

* *iNOS*, inducible nitric oxide synthase; *IL*, interleukin; *MMP*, matrix metalloproteinase; *NFATc1*, nuclear factor of activated T cells cytoplasmic 1; *CtsK*, cathepsin K; *RANKL*, receptor activator of nuclear factor κB ligand; *OPG*, osteoprotegerin; *GAPDH*, glyceraldehyde 3-phosphate dehydrogenase.

## Data Availability

Data are available in a publicly accessible repository.

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
