# Peer review of "Lacticaseibacillus rhamnosus MG4706 Suppresses Periodontitis in Osteoclasts, Inflammation-Inducing Cells, and Ligature-Induced Rats"

_nutrients, 2022, doi:10.3390/nu14224869_

Round 1

Reviewer 1 Report

In manuscript “Lacticaseibacillus rhamnosus MG4706 suppresses periodontitis in osteoclasts, inflammation-inducing cells, and ligature-induced rats”, the authors reported the inhibitory effect of LAB (mainly MG4706) on NO/iNO expression in PgLPS activated RAW264.7 cells and inflammatory markers increase induced by PgLPS in HGH-1 cells. They also reported MG4706 suppressed alveolar bone loss and inflammatory markers decrease in gingival tissues in a periodontitis animal model. These findings are interesting and significant. The manuscript is well written and experimental data are presented in a logical way. 

1) TNF-α and COX2 mRNA level need to be characterized in the cell-based study and rat tissue.

2) PGE2 is an important inflammatory mediator and should be characterized.

3) Pro-inflammatory cytokines concentration should be characterized in the cell-based study.

Author Response

Reviewer 1

Thanks for your attentive comments.

Below is a summary of the answers to each comment.

Point 1: TNF-α and COX2 mRNA levels need to be characterized in the cell-based study and rat tissue.

Response 1: Thank you for the reviewer’s comment. The expression of NO/iNOS is an important factor in periodontitis. Herrera, et al. (2011) suggested that alveolar bone loss was significantly inhibited when iNOS inhibitor was administered to a ligature-induced rat model, and bone resorption activity was also significantly inhibited in the iNOS knockdown model [9]. Therefore, our study was conducted focusing on the expression of NO/iNOS. In addition, the expression of IL-6 and IL-1β is associated with sustained tissue destruction, and these cytokines promote the destruction of alveolar bone and extracellular matrix by periodontitis (Baek, et al., 2013) [10]. Therefore, our study was conducted focusing on IL-6 and IL-1β among pro-inflammatory cytokines.

We supplemented the above-mentioned contents on lines 37-43 in the Introduction section and added the cited references [9, 10] to the Reference section.

Point 2: PGE2 is an important inflammatory mediator and should be characterized.

Response 2: Thanks for the reviewer’s comments. We agree with the reviewer. However, as mentioned in response 1, we determined that NO/iNOS expression or activity was more closely related to periodontitis than PGE2/COX-2 expression, and this study was performed.

Point 3: Pro-inflammatory cytokines concentration should be characterized in the cell-based study.

Response 3: Thanks for the reviewer’s comments. We agree with the reviewer. However, we judged that cellular levels of pro-inflammatory cytokines correlated with mRNA expression. In addition, as presented in this study, it was confirmed that periodontal tissue destruction in an animal model was due to regulating the expression of these cytokines. The exact signaling pathway will be confirmed by protein expression in further animal experiments.

Reviewer 2 Report

Introduction - is very brief, it would be appropriate to add more information related to the topic

Abstract – must be shorter (actual version has 233 words)

Line 20 -21 – the sentence is not suitable for abstract, delete it or add to the introduction

Material and Method

2.5. Quantitative Real Time-Polymerase Chain Reaction (qRT-PCR) – is insufficiently described, a lot of information is missing, e.g. about the temperature program, efficiency..etc – must be added

Results

Line 145 – it is not necessary to state the full name of used the abbreviation (iNOS) several times

Author Response

Reviewer 2

Thanks for your attentive comments.

Below is a summary of the answers to each comment.

Point 1: Introduction - is very brief, it would be appropriate to add more information related to the topic.

Response 1: Thanks for the reviewer’s comments. Accordance with the reviewer's comment, we added relevant content on lines 37-43 and added the cited references [9, 10] to the Reference section.

Lines 37-43: “NO is produced from inducible nitric oxide synthase (iNOS) and causes bone loss by regulating proinflammatory cytokine expression. As mercaptoethyl guanidine, one of the iNOS inhibitors, reduces bone loss, NO is used as a pharmaceutical target in periodontal disease management [9]. The expression of IL-1β and IL-6 is associated with sustained tissue destruction, and these cytokines promote the destruction of alveolar bone and extracellular matrix by periodontitis [10].”

Point 2: Abstract – must be shorter (actual version has 233 words)

Response 2: Thanks for the reviewer’s comments. We shortened the Abstract (184 words) content.

Point 3: Line 20 -21 – the sentence is not suitable for abstract; delete it or add to the introduction.

Response 3: Thanks for the reviewer’s comments. We have deleted the sentence (line 20).

Point 4: Material and Method - 2.5. Quantitative Real Time-Polymerase Chain Reaction (qRT-PCR) – is insufficiently described, a lot of information is missing, e.g. about the temperature program, efficiency..etc – must be added

Response 4: Thanks for the reviewer’s comments. We added the details below on lines 114-116.

“The cDNA was synthesized with an equivalent amount of RNA (1 μg), and PCR was performed on the synthesized cDNA using iQ SYBR Green Supermix (Bio-Rad, Hercules, CA, USA).”

Point 5: Results - Line 145 – it is not necessary to state the full name of used the abbreviation (iNOS) several times.

Response 5: Thanks for the reviewer’s comments. We only marked iNOS on line 148.

Round 2

Reviewer 2 Report

Material and method - 2.5. Quantitative Real Time-Polymerase Chain Reaction (qRT-PCR) - is still insufficiently described, important information is missing, e.g. about the RealTime PCR temperature program, efficiency...etc - need to be added!!, these data are essential and need to be added. Anyway, these are the data that are commonly reported in scientific journals!
